# Interactions between miRNAs and Double-Strand Breaks DNA Repair Genes, Pursuing a Fine-Tuning of Repair

**DOI:** 10.3390/ijms23063231

**Published:** 2022-03-17

**Authors:** Ricardo I. Peraza-Vega, Mahara Valverde, Emilio Rojas

**Affiliations:** Departamento de Medicina Genómica y Toxicología Ambiental, Instituto de Investigaciones Biomédicas, Universidad Nacional Autónoma de México, Ciudad Universitaria, Ciudad de Mexico 04510, Mexico; ricardoivan@gmail.com (R.I.P.-V.); mahara@iibiomedicas.unam.mx (M.V.)

**Keywords:** miRNAs, DNA repair, double-strand breaks, non-homologous end joining, homologous recombination repair

## Abstract

The repair of DNA damage is a crucial process for the correct maintenance of genetic information, thus, allowing the proper functioning of cells. Among the different types of lesions occurring in DNA, double-strand breaks (DSBs) are considered the most harmful type of lesion, which can result in significant loss of genetic information, leading to diseases, such as cancer. DSB repair occurs through two main mechanisms, called non-homologous end joining (NHEJ) and homologous recombination repair (HRR). There is evidence showing that miRNAs play an important role in the regulation of genes acting in NHEJ and HRR mechanisms, either through direct complementary binding to mRNA targets, thus, repressing translation, or by targeting other genes involved in the transcription and activity of DSB repair genes. Therefore, alteration of miRNA expression has an impact on the ability of cells to repair DSBs, which, in turn, affects cancer therapy sensitivity. This latter gives account of the importance of miRNAs as regulators of NHEJ and HRR and places them as a promising target to improve cancer therapy. Here, we review recent reports demonstrating an association between miRNAs and genes involved in NHEJ and HRR. We employed the Web of Science search query TS (“gene official symbol/gene aliases*” AND “miRNA/microRNA/miR-”) and focused on articles published in the last decade, between 2010 and 2021. We also performed a data analysis to represent miRNA–mRNA validated interactions from TarBase v.8, in order to offer an updated overview about the role of miRNAs as regulators of DSB repair.

## 1. Introduction

DNA is continuously exposed to a series of physical and chemical agents that can alter its structure, generate mutations, promote genomic instability and, eventually, cause cell death, which favors the development of diseases, such as cancer. Among the various types of lesions arising in DNA, double-strand breaks (DSBs) are considered highly cytotoxic DNA lesions, since they can lead to an increased rate of mutations, thus, promoting chromosomal aberrations and substantial loss of genetic information, which, ultimately, leads to premature aging, neurodegeneration, immunodeficiency, cancer and cell death, [1,2] There are an estimated ten DSBs per day per dividing mammalian cell [3,4,5,6]. DSBs can be produced by exposure to ionizing radiation (IR), replication errors, and inadvertent cleavage by nuclear enzymes and chemical compounds (i.e., bleomycin, doxorubicin), some of which are employed for cancer treatment.

DSBs are repaired through two main DNA repair mechanisms, called non-homologous end joining (NHEJ) and homologous recombination repair (HRR). The intervention of one mechanism or another depends largely on the phase of the cell cycle in which the lesion has occurred; NHEJ is mainly active in G0/G1 and can be active throughout the cell cycle [5], while HRR predominates in S/G2, when homologous templates are available [6,7]. The NHEJ is considered an error-prone mechanism [3], while HRR is considered an error-free process [8].

The repair of DSBs involves complex signaling events, with a large variety of proteins acting as sensors, transducers and effectors [9]. Accurate regulation of the expression of genes whose proteins are involved in such mechanisms is, therefore, critical to achieve an efficient repair of DNA lesions. In this way, miRNAs [10] could act as a genetic switch by fully down-regulating their targets [11]. However, the accepted notion is that a single miRNA can regulate in a mild degree, hundreds of mRNAs can cooperatively function as micromanagers of genetic expression [12,13]. In this sense, miRNAs act to maintain the average expression level of a gene and to reduce the variation in its expression [11,14].

Modifications in the expression levels of these miRNAs interfere with the activity of the DNA repair mechanism. Interestingly, DNA damage has been shown to promote preferential biogenesis of certain miRNAs [15]. However, the directionality of this regulatory process requires additional analysis.

For the preparation of this review, we employed the following Web of Science search query TS = w(gene name/aliases and (“miRNA” or “miR-” or “microRNA”)). A manual inspection of the search results was performed on recent published papers (2010 to 2020) in which a direct or indirect association between miRNAs and NHEJ/HRR genes had been demonstrated, thus, suggesting a miRNA-mediated modulation of DSBs.

According to a data analysis, performed to obtain validated human miRNA–mRNA interactions, involving the genes reviewed in this work, there are 210 miRNAs that interact with HRR genes and 275 miRNAs interacting with NHEJ genes, thus, forming 362 and 358 interactions in HRR and NHEJ, respectively (Figure 1 and Figure 2). Interactions were obtained from TarBase v.8 database (14 February 2021) [16], which contains experimentally validated miRNA–mRNA interactions, through techniques including microarray, reporter gene assay, qPCR, western blot and RNA-Seq. However, most of these interactions remain to be studied within the context of DNA repair.

## 2. MiRNA-Mediated Regulation of NHEJ

DSBs, occurring throughout the cell cycle, are repaired predominantly by the NHEJ pathway [5]. In human cells, NHEJ repairs nearly all DSBs outside of S and G2 cell cycle phases and even about 80% of DSBs within S and G2 that are not proximal to the replication fork [18,19]. More detailed reviews about DNA repair by NHEJ can be found elsewhere [5,19,20]. There is an increasing number of miRNAs reported to act as regulators of the various genes involved in the NHEJ repair. Disturbances in the expression of such genes could have important consequences for the fate of cells, due to inadequate repair of DSBs, resulting in proliferation perturbation and apoptosis [21]. Below, we describe the relationship between the main genes involved in the DSB repair mechanism and some microRNAs.

## 3. Ku70/80

When a DSB arises in the DNA of mammalian cells, it is first recognized by a heterodimer, consisting of Ku70 and Ku80 (Ku), considered as a Ku:DNA complex, which serves as a node at which the nuclease, polymerases, and ligase of NHEJ can dock later (Figure 3) [3,18].

There are several miRNAs reported to modulate Ku70 (*XRCC6*) and Ku80 (*XRCC5*) expression (Figure 3). It has been observed that miR-545 increases radiosensitivity in lung carcinoma cells via inhibiting Ku70 expression [22]. Similarly, up-regulation of miR-379-5p inhibited proliferation and sensitized granulosa cells to DNA damage by repressing Ku70 [23]. MiR-124 has been shown to be up-regulated in brain tissues of ischemia stroke mice and rats. The targeting of Ku70 by miR-124 under these conditions, has been associated with altered cell proliferation and apoptosis [24]. Transfection with miR-890 caused a reduced expression of Ku80, which resulted in a significantly delayed IR-induced DNA damage repair in a prostate cancer model [25]. It has been identified that miR-623 binds to Ku80, thus, significantly decreasing its expression, causing a reduction in cell proliferation, clonogenicity, migration and invasion in lung adenocarcinoma cells [26]. Low constitutive Ku80 mRNA expression, together with radiation-induced high miR-99a expression in peripheral blood lymphocytes, were associated with late rectal bleeding in prostate cancer patients, receiving intensity-moderated radiation therapy [27]. Further, miR-31 was shown to be up-regulated in squamous cell carcinoma and it has been identified to target Ku80, promoting its down-regulation and causing impairment of DNA repair activity [28]. It has been identified that miR-526b binds to Ku80 mRNA, thus, modulating Ku80 expression in non-small cell lung carcinoma cells. miR-526b overexpression was shown to induce S-phase cell cycle arrest and apoptosis and enhanced the expression levels of the p53 and p21, thus, inhibiting lung tumor growth [29]. It has been observed that miR-622 regulates the expression of the Ku complex through direct interaction with Ku70 and Ku80 transcripts, and specifically suppresses NHEJ during the S cell cycle phase and enhances initiation of HR-mediated DSB repair in the S phase, by facilitating the recruitment of *MRE11*. This latter suggests a role for miR-622 in regulating the balance between HR and NHEJ in the cell cycle [30]. It was demonstrated that miR-502 directly targets Ku70 and XLF in pancreatic cell lines, which might modulate both the assembly of the Ku70/80 complex and final steps of ligation during NHEJ; overexpression of this miRNA makes cells susceptible to DNA damage [31].

## 4. DNA-PKcs

Ku functions as a specific cofactor for stable recruitment of the DNA-protein kinase catalytic subunit DNA-PKcs (encoded by *PRKDC**/XRCC7*) to DSB sites (Figure 3). DNA-PKcs, thus, acts as a sensor for DSBs, and its major role is to promote NHEJ [32]. The kinase activity of DNA-PKcs regulates end processing and NHEJ through autophosphorylation and also facilitates recruitment of downstream effectors. Given the role of DNA-PKcs in DSB repair, it becomes evident that alterations in its expression might compromise the activity of NHEJ downstream effectors, thereby affecting the ability of cells to deal with DNA damage. There are a number miRNAs identified to target DNA-PKcs (Figure 3). It has been reported that miR-101 directly targets *PRKDC*. Up-regulation of the miRNA efficiently reduced the protein levels of DNA-PKcs in human lung cancer cells and sensitized them to radiation, both in vitro and in vivo [33,34]. Sensitization to gemcitabine was also observed in pancreatic cancer cells [35] and there is an augmented salinomycin cytotoxicity against osteosarcoma cells [36]. As expected, down-regulation of miR-101 has been associated with DNA-PKcs overexpression and mediated oncogenic actions in renal and hepatocellular carcinoma cells [36,37]. Likewise, ectopic expression of miR-488 resulted in markedly increased cisplatin sensitivity of malignant melanoma cells, due to DNA-PKcs silencing, in vitro and in vivo [38,39] (Figure 3). miR-136 overexpression down-regulated DNA-PK in ovarian cancer cells [40]. It has been shown that overexpression of miR-874-3p has a great impact on the survival of non-small cells lung cancer (NSCLC) A549 cells after irradiation, by targeting *DNA-PKcs* and, therefore, leading to a 20% radiosensitivity increase [41]. miR-1323 was demonstrated to bind to *DNA-PKcs* in A549 cancer cells. Ectopic expression of this miR-1323 significantly promoted DNA-PKcs expression and increased the survival of irradiated cells. This latter suggests a plausible mechanism of resistance to radiation via enhanced DNA repair [38].

It has been observed that miR-3162 forms a miRNA/Ago2/YY1/PcG group protein/DNMT complex that can bind to the promoter region of DNA-PKcs, contributing to histone and DNA methylation in bortezomib-treated leukemic cells, thus, mediating the epigenetic gene silencing of DNA-PKcs [42].

## 5. DCLRE1C, DNA Pol M and Pol L

Some DSBs that are repaired by NHEJ require additional accessory factors, such as nucleases for adequate repair. One such factor is the endonuclease DCLRE1C (Artemis), which interacts with DNA-PKcs to promote the repair of a subset of DSBs, which appear to represent those that incur some level of resection, possibly owing to their increased complexity [32,43] (Figure 3). DNA *POLM* (Pol µ) and *POLL* (Pol λ) are two low-fidelity polymerases, involved in NHEJ in human cells [44,45]. Both polymerases can incorporate nucleotides in a template-dependent or template-independent manner [46]. The activity of these polymerases further explains the high level of diversity that can occur at NHEJ junctions and demonstrated that, although resection is one way of generating short stretches of homology between broken DNA ends, the template-independent nucleotide addition of one or both broken DNA ends is another [18]. At the time of writing this review, there are still no published papers about a miRNA-mediated regulation Artemis, Pol μ and Pol λ.

## 6. XRCC4: DNA Ligase IV Complex and XLF

DNA ligase IV (LIG4) is the most flexible known ligase, since it has the ability to ligate across gaps and ligate incompatible DNA ends [3,47] (Figure 3). LIG4 functions exclusively in NHEJ, making it a central component for the DNA repair process. The protein encoded by *XRCC4* combines with LIG4 in the repair of DSBs through NHEJ. Coupling with *XRCC4* stimulates LIG4 enzyme activity and loss of either LIG4 or XRCC4, and severely compromises NHEJ [18]. *XRCC4* is a direct target of miR-151a, and it has been observed that stable overexpression of miR-151a recovered the temozolomide (TMZ) sensitivity of TMZ-resistant glioblastoma (GBM) cells in vitro, thus, demonstrating the importance of miR-151a/XRCC4 signaling in modulating NHEJ repair and TMZ resistance in GBM [48]. It was demonstrated that miR-1246 packaged in exosomes secreted from irradiated bronchial epithelial BEP2D cells down-regulates LIG4 expression and inhibits proliferation of non-irradiated cells [49]. LIG4 expression was significantly inhibited by overexpression of miR-1587, thus, leading to increased radiosensitivity of CRC cells through DSBs accumulation, cell cycle arrest and apoptosis [50]. XLF is a protein that permits efficient ends ligation by interacting with XRCC4, allowing XLF to complex with XRCC: LIG4 [18]. It remains to be established if there is an XLF regulation mediated by miRNAs.

Vast evidence shows that miRNAs play an important role in the regulation of NHEJ (Figure 3). As we have seen, alterations in the expression levels of NHEJ gene-targeting miRNAs can have an important impact on the DNA damage repair ability of cells, thus, affecting processes, such as radiation and drug sensitivity. However, it remains to be studied in more detail, the processes triggering such alterations, as well as their possible consequences for the cell.

## 7. MiRNA-Mediated Regulation of HRR

HRR requires the invasion of homologous or sister chromatid strands to repair DNA end DSBs, and is considered a typically error-free DNA repair mechanism [51] (Figure 4). A critical initial step in establishing this repair mechanism is DNA end resection, which generates a long 3′ single-stranded DNA that can invade the homologous DNA strand. This step, in addition to blocking the entry of repair Ku proteins by NHEJ, promotes ATM and ATR activation and is, therefore, restricted to the late S phase and G2 phases of the cell cycle [51] (Figure 4). DNA end resection initiation begins with MRN complex (*MRE11*-*RAD50*-NBS1), SRCAP, and CtIP. It is important to mention that the dual activity of *MRE11*, functioning as both endo and exonuclease, is essential to carry out resection initiation. After that, resection extension continues, in which SMARCAD1 cooperates with EXO1 and BLM/DNA2. Subsequently, the process continues when the ssDNA-ends are covered with RPA. In a third step, RPA is replaced by RAD51 in a BRCA1/2-dependent process, to ultimately perform the recombinase reaction, using a homologous DNA template [52]. For a detailed description of HRR, there are more extensive reviews in this regard [8,52,53,54,55].

## 8. *MRE11*-*RAD50*-NBS1 Complex

The complex of proteins *MRE11*-*RAD50*-NBS1 (NBN) (MRN complex) acts as DSB sensor, co-activator of DSB-induced cell cycle checkpoint signaling, and as a DSB repair effector in the HRR pathway [56,57] (Figure 4). A seven-miRNA signature (miR-103, miR-494, miR-99b, miR-21, miR-224, miR-92a, and let-7a) has been reported to be up-regulated in endothelial cells after exposure to radiation, hydrogen peroxide, and cisplatin [58]. Among these miRNAs, miR-494 has been demonstrated to target the MRN complex, which, in turn, leads to senescence and decreased angiogenesis [59].

*MRE11* acts as an exonuclease, generating 3′ overhangs that become a substrate for HRR [60]. *MRE11* overexpression, commonly observed among cancer patients, has been postulated as a mechanism responsible for increasing cancer risk [61]. miR-493-5p has been demonstrated to downregulate *MRE11,* along with 14 other genes involved in genomic stability in *BRCA2* mutant cells, thus, suggesting a possible role for miR-493-5p in regulating DSB repair, through HRR and resistance to *PARP* inhibitors and platinum [62]. It has been observed that miRNAs whose expression levels decrease during aging, when transfected into aged mouse livers, resulted in a significant increase in *MRE11* expression [63]. Additionally, polymorphisms in predicted miRNA target sites of *MRE11, NBS1, RAD51* and *RAD52* have been associated with different cancer risks, susceptibility, and survival [64,65]. It has been reported that miR-153 can bind to *MRE11,* thus, inhibiting its expression in muscle-invasive bladder tumor cells, suggesting a post-transcriptional down-regulation of *MRE11A* due to miR-153 up-regulation [66].

*MRE11* and *RAD50* have been observed to form a large ATP-controlled molecular clamp, suited to recognize even blocked DSBs [67]. An inverse correlation between miR-183 and *RAD50* expression levels has been observed after pomegranate extract treatment in MCF-7 cells, accompanied by an increase in DSBs, as revealed by an elevated frequency of γ-H2AX foci, followed by cell cycle arrest in G2/M and the induction of apoptosis [68].

It was found that *Staphylococcus aureus* induces host miR-15b-5p, in both acute wounds and diabetic foot ulcers (DFUs). *S. aureus*-mediated overexpression of miR-15b-5p resulted in suppression of DNA damage repair mechanisms and inflammatory response in DFUs. Putative target genes of miR-15b included *RAD50*, which was down-regulated in DFUs. Because of the inhibition of DNA repair mechanisms, DSBs were accumulated in DFU tissue, as indicated by the presence of γH2AX, indicating a mechanism of *S. aureus*-mediated induction of miR-15b that results in deregulation of DNA repair mechanisms and inflammatory response, contributing to the inhibition of healing in DFUs [69].

Further, miR-24 caused a decrease in the expression of *NBS1* by targeting c-Myc, which is known to positively regulate the expression of *NBS1*. Down-regulation of *NBS1* by miR-24 was thought to improve myoblasts’ permissiveness to adeno-associated virus (AAV) transduction, possibly by avoiding MRN sensing and binding to AAV genome [70].

## 9. CtIP, EXO1 and RPA

In mammals, DNA-end resection is catalyzed by the MRN complex, together with CtIP (CtBP-interacting protein/RBBP8) and EXO1 [71] (Figure 4). CtIP (*RBBP8*) physically interacts with MRN and, along with EXO1, promotes DSB resection [72]. The targeting of CtIP by aberrantly expressed miR-19 impairs CtIP-mediated DNA-end resection, which resulted in reduced HRR levels and DNA damage hypersensitivity [73]. IR-induced DSBs activate DSB signaling kinase ATM, which leads to the downregulation of miR-335. Overexpression of miR-335 in HeLa cells resulted in reduced CtIP levels and low post-IR colony survival and BRCA1 foci formation. Further, in two patient-derived lymphoblastoid cell lines, with decreased post-IR colony survival, a radiosensitive phenotype, it was demonstrated that elevated miR-335 expression reduced CtIP levels and reduced BRCA1 foci formation [74]. It was demonstrated that miR-18a-5p directly binds to CtIP and inhibits its expression in nasopharyngeal carcinoma cells, thus, affecting cell proliferation and apoptosis. However, miR-18a-5p activity on CtIP is modulated by the lncRNA CASC2 via sponging miR-18a-5p and modulating *CtIP* expression in vivo [75]. It has been observed that *CtIP* is a miR-19 target and aberrant expression of the miRNA suppresses DNA-end resection and faithful repair of DSBs by HRR in HEK293T cells, thus, promoting genomic instability [75]. Overexpression of miR-130b resulted in increased DNA damage levels by promoting DSBs through CtIP targeting, as evidenced by elevated phosphorylation of H2AX and increased DNA damage, observed through comet assay in Hela and Siha cells, which was accompanied by accelerated cell apoptosis in, combination with PARP inhibitors [76].

RPA is a heterotrimeric complex (RPA1/2/3), which binds to single-stranded DNA, forming a nucleoprotein complex, protecting from nucleases, preventing formation of secondary structures that would interfere with repair [77,78]. It was reported that miR-519 repressed the expression of *RPA* and *EXO1,* thus, augmenting DNA damage, as revealed by the increase in γH2AX foci in miR-519-transfected HeLa cells [79]. Lastly, it was observed that miR-30 binds directly to *RPA*, overexpression of the miRNA increased DNA damage, cell cycle arrest and inhibited cell proliferation in gastric cancer cells [80]. In another study, miR-493-5p has been demonstrated to downregulate RPA, along with 14 other genes involved in genomic stability in BRCA2 mutant cells, thus, suggesting a possible role for miR-493-5p in regulating DSB repair, through HRR and resistance acquisition of BRCA2 mutant cells to PARP inhibitors and platinum [62].

## 10. BRCA1 and BRCA2

BRCA1 and BRCA2 are important proteins for the repair of DSBs by HRR. There are numerous reports demonstrating that BRCA1 and BRCA2 expression can be modulated, either directly or indirectly by miRNAs. It has been established that miR-1255b, miR-148b and miR-193b directly target the transcripts of BRCA1, BRCA2 and *RAD51,* therefore, suppressing the HRR pathway during the G1 phase. Inhibition of these miRNAs increases BRCA1, BRCA2 and *RAD51* expression in the G1 phase, thus, leading to impaired DSB repair, which, in turn, can lead to a loss of heterozygosity [81]. BRCA1/BRCA2 mutations and aberrant expression are common as the hallmarks of ovarian and breast cancer [82,83]. It has been reported that miR-182 overexpression contributes to aggressive ovarian cancer, largely by its negative regulation of multiple tumor suppressor genes, involved in tumor growth, invasion, metastasis, and DNA instability [84]. It was reported that expression of BRCA1 and BRCA2, both let-7a targets, was down-regulated after delivery of magnetic core–shell nanoparticle (MCNP)/let-7a constructs to breast cancer cells. Co-delivery of let-7a with doxorubicin induced a synergistic effect, which resulted in a significant diminution of chemoresistance, DNA repair and cell viability [85]. Up-regulation of miR-146a, -148a and -545 were associated with overall survival and progression-free survival in patients with wild-type BRCA1/2. These miRNAs were predicted to target BRCA1/2 and, thus, inferred to alter the DNA damage repair [86]. In a screen to evaluate tumor suppressor genes, oncogenes and miRNA expression in samples of colorectal cancer, it was found that miR-17, miR-425 and miR-92 overexpression were significantly associated with up-regulated expression of BRCA1, suggesting a possible role of these miRNAs in the development of the disease [87].

BRCA1 forms a complex with CtIP and the MRN complex and, thus, plays a crucial role in promoting DSB end resection, which generates a long 3′ single-stranded DNA tail that can invade the homologous DNA strand [88,89]. MiR-182, a BRCA1 negative regulator, is significantly overexpressed in high-grade serous ovarian carcinoma, which results in a significant reduction in BRCA1 expression. As a consequence, this can cause impairment of the repair of DSBs, increased tumor transformation, invasiveness and metastasis [90,91]. Additionally, it has been observed that overexpression of miR-182 impairs HRR activity and renders HL60 cells hypersensitive to IR [92]. Furthermore, it has been proposed that TGFβ acts as stringent regulator of the DNA damage response via suppression of miR-182, which directly targets BRCA1 [93]. In breast tumors cell lines, high levels of miR-146a and miR-146b were inversely correlated with BRCA1 expression. Such direct downregulation was associated with an increased proliferation and reduced HRR rate [94,95].

In a screen to determine miRNAs modulating cell response to DNA damage, it was found that the expression of members of the miR-99 family were up-regulated, following DNA damage, and miR-99 expression correlated with radiation sensitivity. The downregulation of the remodeling complex SNF2H, a miR-99 family target, mediated IR sensitivity through its role in facilitating DNA repair. The up-regulation of the miR-99 family, following IR, decreased the efficiency of BRCA1 recruitment and the rate of DNA repair after exposure to IR [96]. In a different study, it was demonstrated that miR-210-3p attenuates the G2/M cell cycle checkpoint by inactivating the BRCA1 complex function, in response to DNA damage under hypoxia, via targeting the 3′ UTR region of *BARD1* mRNA in endometrial stromal cells [97].

The overexpression of miR-7 caused indirect down-regulation of BRCA1 in lymphoblasts, accompanied by an increase in γH2AX, reflecting augmented DNA damage, apoptosis and proliferation inhibition [98]. MiR-212 was identified to contribute to radio-resistance of U251 glioma cells and was shown to attenuate radiation-induced apoptosis by BRCA1 direct targeting [99]. It was confirmed that miR-9 binds directly to BRCA1, thus, decreasing gene expression. Such low BRCA1 expression and high expression of miRNA-9 was associated with platinum sensitivity and longer progression-free survival in ovarian cancer cells [100]. Down-regulation of miR-218 was observed in cisplatin-resistant breast cancer cell lines and it was identified that BRCA1 was the cellular target of miR-218. Restoring miR-218 expression in the MCF-7/DDP cell line could sensitize cells against cisplatin, thereby increasing cisplatin-mediated tumor cell apoptosis and reducing DNA repair [101]. In another study, miR-638 overexpression increased sensitivity to DNA-damaging agents, UV and cisplatin, reduced proliferation rate, as well as decreased invasive ability in TNBC cells. Reduced proliferation, invasive ability, and DNA repair capabilities were associated with down-regulated BRCA1 expression [102].

Overexpression of miR-335 in MCF7 cells resulted in an up-regulation of BRCA1 expression, through inhibition of BRCA1 negative regulator, ID4, which, in turn, impacts on cellular functions, such as proliferation and apoptosis [103]. Expression of miRNA-26a, -29b, -100, and -148a increased in BRCA1 wild type cells (MDB-MB-231), after exposure to gemcitabine, alone and in combination with the PARP1 inhibitor [104]. miR-498 has been found highly expressed in triple-negative breast cancer cells, and its expression was negatively correlated with the levels of BRCA1. It was demonstrated that miR-498 inhibits BRCA1 expression, thus, leading to promoted cell proliferation [105].

BRCA1 was also reported to be targeted by miR-7-5p in non-small cell lung cancer cells. It was observed that such BRCA1 targeting is regulated through lncRNA MEG3 by competitive binding to miR-7-5p [106]. miR-7-5p overexpression was also associated with suppression of cell proliferation and promotion of apoptosis by targeting BRCA1 in TK6 cells [98]. It was reported that BRCA1 can also act as a transcriptional repressor of miR-155 [107]. Up-regulation of miR-155 in breast cancer is likely related to tumor initiation. In contrast to BRCA1, *FOXP3* is a transcriptional inducer of miR-155 in breast cancer cell lines. miR-155 is associated with transcriptional regulation by *FOXP3,* through BRCA1 transcriptional inhibition, subsequently controlling miR-155 and its targets, such as RAD51 [107]. MiR223-3p is a negative regulator of the NHEJ DNA repair and represents a therapeutic pathway for BRCA1- or BAP1-deficient cancers [108].

BRCA2 is targeted by miR-19a-3p in NSCLC, A549; the overexpression of this miRNA affects cell survival after irradiation [41]. BRCA2 is the main mediator of RAD51 nucleofilament formation and strand exchange in mammalian cells [51]. Mutations affecting the proper function of these proteins are associated with increased genomic instability. High genomic instability is thought to be responsible for the significantly increased cancer risk in patients with familial or germline BRCA mutations [109].

## 11. RAD51

RAD51 forms nucleoprotein filaments with DNA catalyzing the transfer of the new DNA strand between a broken sequence and its homolog, to resynthesize the damaged region [55]. Transfection with let-7e sensitized epithelial ovarian cancer cells to cisplatin, down-regulated BRCA1 and *RAD51* expression, and repressed the repair of cisplatin-induced DSBs [110]. It has been reported that, miR-182 targets *RAD51* and over-expression of the miRNA decreases HRR, thus, sensitizing acute myelogenous leukemia cells to capecitabine [111]. Overexpression of miR-155 in human breast cancer cells reduces the levels of RAD51 and affects the cellular response to IR, decreasing the efficiency of HRR and enhancing sensitivity to IR [112]. Additionally, HRR decreased efficiency by miR-155 overexpression drives a concurrent NHEJ activity increase and, thus, a higher mutation frequency [113]. Over-expression of miR-203 also increased IR sensitivity of human malignant glioma cell lines and prolonged radiation-induced γH2AX foci formation. It was demonstrated that miR-203 down-regulated *RAD51* among other DNA damage response genes, which resulted in inhibited invasion and migration potentials [114]. It was observed that miR-34a directly binds and modulates *RAD51* expression, therefore, inhibiting IR-induced DSBs repair in non-small cell lung cancer. Liposomal formulation, containing miR-34a, MRX34 plus radiotherapy, effectively inhibits tumor growth in lung cancer mouse models [115]. Down-regulation of *RAD51* by miR-506 affects the HRR pathway and, consequently, increases sensitivity to cisplatin and the PARP inhibitor olaparib in serous ovarian cancer cells [116,117]. In a similar fashion, miR-103 and miR-107, when overexpressed, reduced HRR and sensitized osteosarcoma cells to various DNA-damaging agents, including cisplatin and a PARP inhibitor by targeting *RAD51* [118]. Likewise, miR-107 and miR-222 regulate the DNA damage repair and sensitize tumor cells to olaparib by repressing expression of *RAD51*, thus, impairing DSB repair by HRR [119]. Another study showed that miR-96 directly targets the coding region of *RAD51*. Overexpression of miR-96 decreased the efficiency of HRR and enhanced the sensitivity of osteosarcoma cells to the PARP inhibitor, AZD2281, and cisplatin, indicating that miR-96 regulates DNA repair and chemosensitivity by repressing *RAD51* [120]. Ectopic expression of miR-152 reversed cisplatin resistance, both in vitro and in vivo, by targeting *RAD51*. On the other hand, it was demonstrated that miR-98-5p could directly target Dicer, causing global miRNA down-regulation. However, miR-152 was identified as the main downstream target of miR-98-5p. Importantly, miR-98-5p expression was associated with poor outcome of epithelial ovarian cancer patients, presumably by indirectly regulating the biogenesis of miR-152 [121]. It has been shown that miR-193a-3p binds to *RAD51* and modulates its expression. Such interaction is, in turn, modulated by the long RNA *lnc-RI,* which is also recognized by miR-193a-3p and acts as a competitive endogenous RNA (ceRNA) to stabilize *RAD51* mRNA via competitive binding with miR-193a-3p [122].

## 12. Pol δ

Extensive evidence shows that POL δ (POLD) is central to HRR-associated DNA synthesis. POLD possesses both high fidelity polymerase and 3′ to 5′ exonuclease activity. This polymerase can efficiently extend up to 80% of the RAD51-mediated invading DNA strands [123]. It has been reported that miR-155 could modulate *POLD1* expression through the targeting of *FOXO3a*, a transcription factor with putative binding sites in the promoter of each of the four polymerase delta subunits [113].

## 13. BLM and LIG1

BLM is a member of the RecQ family of DNA helicases, and its role in HRR is related to the dissolution of Holliday junctions, generated during the strand invasion step. Mutations in BLM are present in Bloom syndrome patients, and this syndrome is characterized by chromosomal instability and a 10-fold elevation in the frequency of sister chromatid exchanges [124]. BLM expression has been demonstrated to be regulated by miR-493-5p, with this miRNA also regulating the expression of 14 other genes involved in genomic stability in BRCA2 mutant cells, thus, suggesting a possible role for miR-493-5p in regulating DSB repair through HRR and resistance acquisition of BRCA2 mutant cells to PARP inhibitors and platinum [62].

The DNA synthesis from the 3′-end of the invading strand is followed by successive ligation by DNA ligase I (*LIG1*). A study indicates that *LIG1* targeting by miR-874 promotes a radiosensitizing effect in lung cancer cells [41]. So far, no other miRNAs regulating the expression of LIG1 and its consequences on DNA repair have been reported.

## 14. Other Validated MiRNA–MRNA Interactions

There are a large number of miRNA–mRNA interactions that remain to be studied, to better understand the role of miRNAs in the repair of DSBs. According to a data analysis performed to obtain validated human miRNA–mRNA interactions, involving the genes reviewed in this work, there are 210 miRNAs that interact with HRR genes and 275 miRNAs interacting with NHEJ genes, thus, forming 362 and 358 interactions in HRR and NHEJ, respectively (Figure 1 and Figure 2). Interactions were obtained from TarBase v.8 database [16], which contains experimentally validated miRNA–mRNA interactions, through techniques including microarray, reporter gene assay, qPCR, western blot and RNA-Seq. The vast majority of these interactions remain to be studied within the context of DNA repair. Interestingly, *PRKDC* mRNA (DNA*-PKcs*) interacts with 163 different miRNAs, which places it as the gene with the highest number of validated interactions in NHEJ (Appendix A). The functioning of NHEJ depends largely on the activity of DNA-PKcs; therefore, its expression needs to be highly regulated so there is an adequate repair of DSBs. Such a degree of regulation could be mediated by the large number of miRNAs targeting *PRKDC*. Regarding the HRR genes, *NBN, EXO1*, and BRCA1 top the list of genes with the most miRNA interactions, with 44, 42 and 41 interactions, respectively (Figure 2; Appendix A). Despite the large number of miRNAs targeting these genes, with the exception of BRCA1, there are very few published studies that explore its regulation mediated by miRNAs.

The miRNA–mRNA interaction data analysis also revealed that miR-155-5p and miR-16-5p are the miRNAs with the most NHEJ gene interactions, by targeting five genes each (*PRKDC, XRCC5,* LIG4, *POLM* and DCLRE1C for miR-155-5p and *XRCC4* for miR-16-5p) (Figure 1; Appendix A). Interestingly, miR-1-3p showed a strikingly large number of interactions with HRR genes. It was observed that miR-1-3p interacts with 11 HRR genes (*BRCA1, BRCA2, EXO1, LIG1, MRE11A, NBN, POLD1, RAD50, RAD51, RBBP8 and RPA1*) (Figure 2; Appendix A). These data suggest a prominent role for these miRNAs in the post-transcriptional regulation of NHEJ and HRR genes; however, their participation in this aspect has not yet been sufficiently studied.

## 15. Concluding Remarks

In recent years, interest and reports about the regulatory role of miRNAs in DNA repair have been increasing. Here we reviewed several examples of miRNAs regulating expression of genes involved in DSBs repair. Since DNA repair mechanisms are of great interest in cancer research, it is not surprising that some of the most studied genes for cancer treatment, such as BRCA1 and RAD51, have an equally high number of reports showing miRNA-mediated regulation of these genes. A deeper study on the functioning of other genes involved in HRR and NHEJ is needed to establish their usefulness as new cancer therapy targets. In this sense, knowing the regulation that miRNAs exert on their expression and activity will be of great importance.

DNA repair mechanisms are a promising target for therapeutic intervention. Many of the anti-cancer therapies are aimed at generating DNA damage, specifically DSBs, thus, reducing cell proliferation and inducing apoptosis. However, development of resistance to radiotherapy, chemotherapy or other targeted therapies can occur during the course of treatment due to deregulation of gene expression. For example, more than one hundred genes have been classified into seven categories of mechanisms of chemoresistance (MOC1-7) in hepatocellular carcinoma (HCC), affecting the response to pharmacological strategies employed for HCC treatment [125]. Deregulation of the genes involved in DSB repair, such as *XRCC4*, *XLF* and *ATM,* have been classified as key players in DNA repair–MOC for HRCC (MOC4) [125]. As has been discussed throughout this review, miRNA-mediated modulation of DNA repair genes can prevent or reverse resistance to chemotherapeutics. Therefore, a better understanding about the regulation of DSB repair by miRNAs will allow us to discover new targets and to develop better strategies to overcome tumor resistance. While it is true that much remains to be learned about the regulation of gene expression through miRNAs, in recent years, there has been a growing interest to study the role of miRNAs in DNA repair and as targets for cancer therapy.

## Figures and Tables

**Figure 1 ijms-23-03231-f001:**
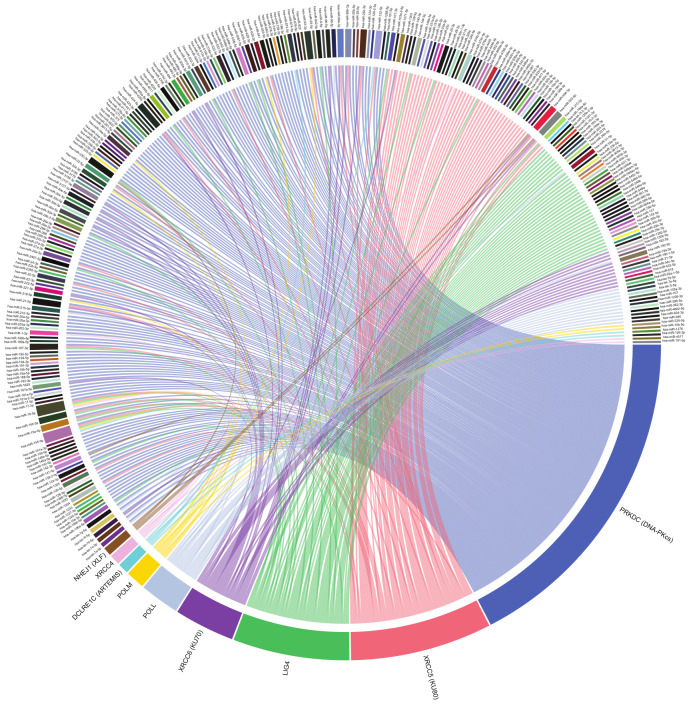
Human miRNA–mRNA-validated interactions involving NHEJ genes according to a data analysis from TarBase v.8 interactions database. There are 275 miRNAs and 358 interactions represented. Graphic created using circlize R package [17].

**Figure 2 ijms-23-03231-f002:**
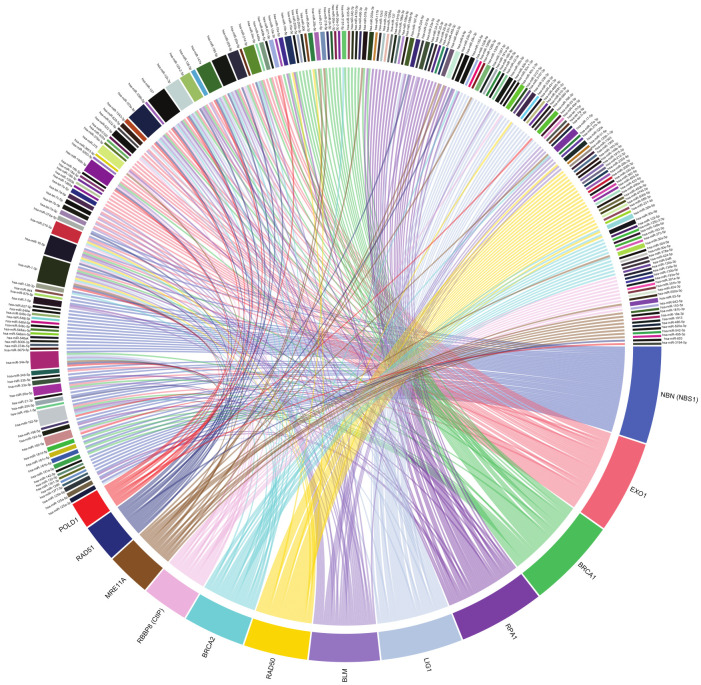
Human miRNA–mRNA interactions involving HRR genes according to a data analysis from TarBase v.8 interactions database. There are 210 miRNAs and 362 interactions represented. Graphic created using circlize R package [17].

**Figure 3 ijms-23-03231-f003:**
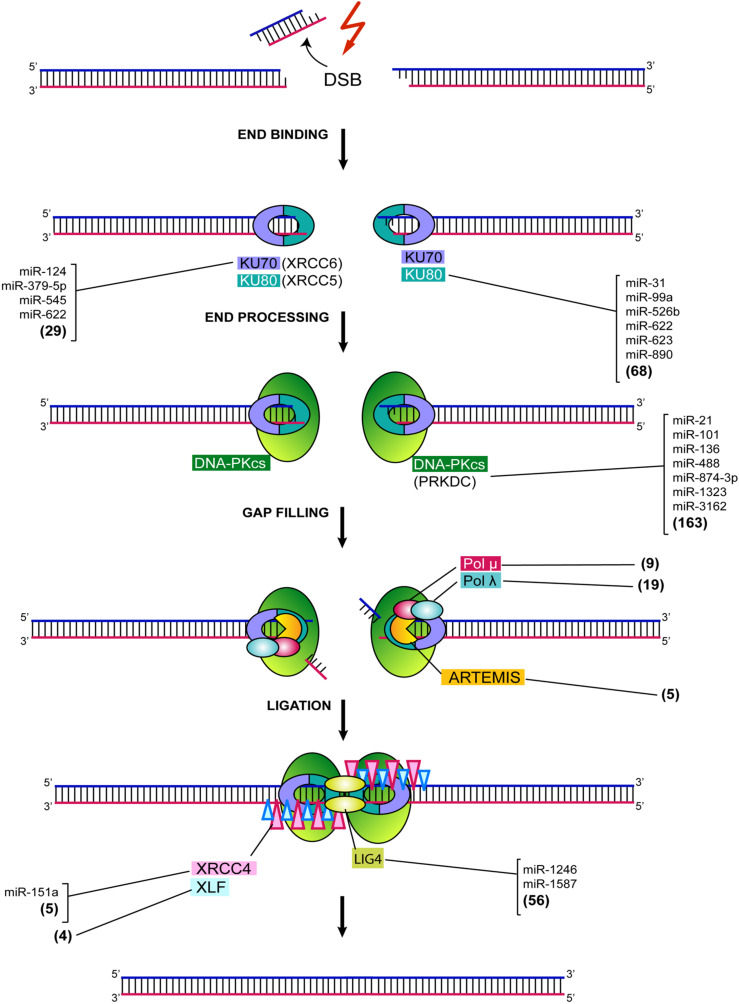
General mechanism for repairing DSBs through NHEJ and miRNAs reported to directly or indirectly modulate gene expression of various NHEJ genes. The NHEJ repair mechanism is presented in stages. DNA complex formation stage, of each Ku protein, value in parentheses refers to the number of miRNA–mRNA interactions validated according to the analysis of the TarBase v.8 interactions database, in the same way for the entire figure. The recruitment of DNA-PKcs and the DSB signaling step show the interactions of *PRKDC* and miRNAs. In the End resection and generation of microhomology regions stage, the number of validated interactions for Pol μ, Pol λ and Artemis is shown. Finally, for the End ligation stage, the specific interactions between XRCC4, XLF and LIG4 are shown through the lines.

**Figure 4 ijms-23-03231-f004:**
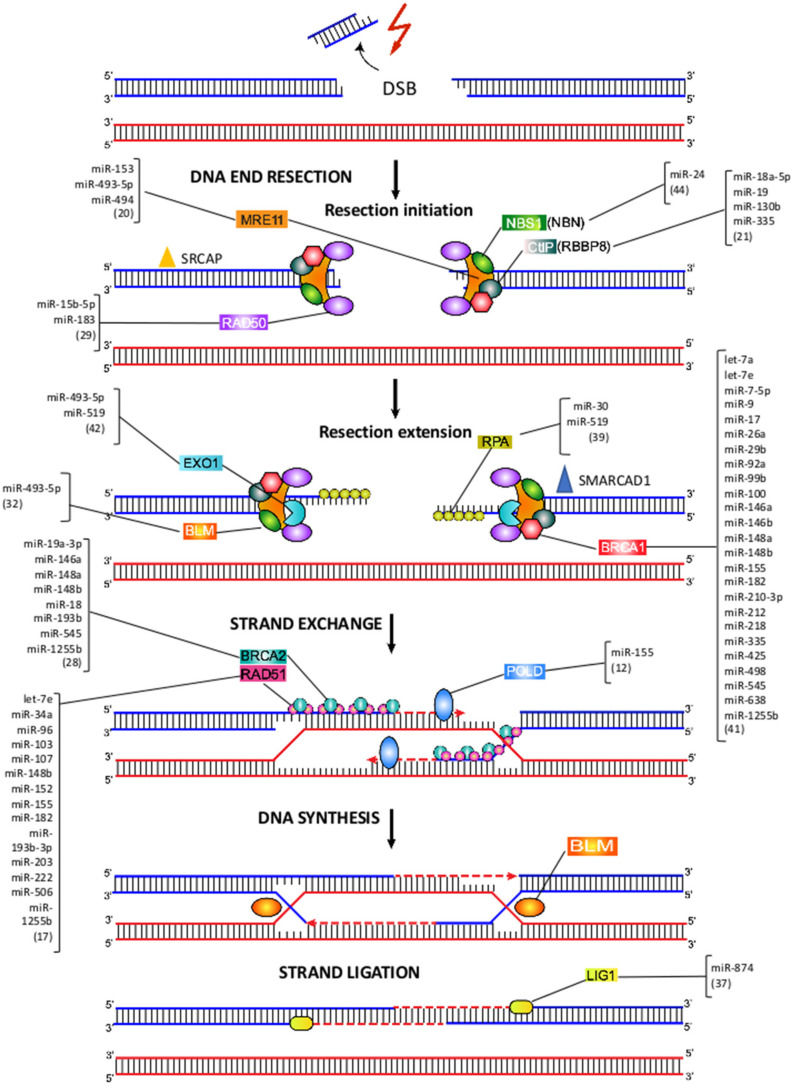
Mechanism of DSB repair by HRR and the miRNAs involved in direct or indirect regulation of some HRR gene expression. The HRR mechanism is represented by stages, the value in parenthesis refers to the number of miRNA–mRNA interactions according to analysis from TarBase v.8 interaction database. In the DSB detection, resection initiation and signaling stage show interactions for *MRE11*, *RAD50*, *NBS1*, and CtIP. Resection extension stage shows interactions of RPA, BRCA1, BLM and EXO1. In homolog strand exchange and DNA synthesis stage is shown miRNA interactions with RAD51, BRCA2 and POLD. Finally for the Strand ligation stage is shown the interaction between LIG1 and miR-874.

## Data Availability

All data used are reflected in the Appendix A.

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
