# Peer review of "Interactions between miRNAs and Double-Strand Breaks DNA Repair Genes, Pursuing a Fine-Tuning of Repair"

_ijms, 2022, doi:10.3390/ijms23063231_

Round 1

Reviewer 1 Report

Ivan et al. submit  to IJMS journal the  review “Interactions between miRNAs and double strand breaks DNA repair genes, pursuing a fine-tuning of repair”. This review is interesting, but requires a revision.

  1. Manuscript requires a list of abbreviations. this is especially true of the name of the genes. Without it, the text is very difficult to understand.
  2. Then, for some proteins involved in DNA repair, the role of miRNAs in regulating their expression is described. It should be explained why these proteins were selected, others were omitted, e.g. Artemis or the multifunctional p53 protein involved in DNA repair processes.
  3. I do not understand why this article was limited to the last 10 years. There are many important older papers concerning DNA repair which were cited thousands time . They should be cited here.
  4. Following the guidelines for authors, citation should be as follows: 1. Author 1, A.B.; Author 2, Title of the article, Abbreviated Journal Name, Year, Volume, page range. Why did the authors of the review change that.

Author Response

Reviewer 1

Dear reviewer, we appreciate your time and comments on the manuscript submitted for publication in Int. J. Mol. science Without a doubt, this will contribute to improve the submitted review work. We know that there is valuable literature that was not considered for this review. However, we would like you to understand that the reason for this is supported by the search criteria established to be able to consolidate information on the close relationship of gene regulation mediated by miRNAs and experimentally validated.

Then, one by one, we address the comments to the work submitted for publication.

  1. Manuscript requires a list of abbreviations. this is especially true of the name of the genes. Without it, the text is very difficult to understand.

A: We agree with your comment and that is why this version includes a list of abbreviations at the beginning of the manuscript.

  1. Then, for some proteins involved in DNA repair, the role of miRNAs in regulating their expression is described. It should be explained why these proteins were selected, others were omitted, e.g. Artemis or the multifunctional p53 protein involved in DNA repair processes.

A: The study approach was limited only to the level of gene expression, in the section MIRNA – MEDIATED REGULATION OF NHEJ this is what we expressed "Disturbances in the expression of such genes could have important consequences on the fate of cells due to inadequate repair of DSBs, resulting in proliferation perturbation and apoptosis [20]. Below we describe the relationship between the main genes involved in DSB repair mechanism and some microRNAs". Thus, the genes included are those in which there is experimental evidence of being regulated by miRNAs and that act specifically in the repair mechanisms of NHEJ and HR. That is, those that resulted from the established search and its terms. We agree that there are many more proteins that play important roles in DNA repair processes; however, we had to narrow the focus of the review.

  1. I do not understand why this article was limited to the last 10 years. There are many important older papers concerning DNA repair which were cited thousands time. They should be cited here. Therefore, from the abstract we describe the parameters established as selection criteria.

A: We agree that there are many articles published over 10 years ago that are relevant in the area of DNA repair; We regret not including them to limit the number of bibliographical references. The references included respond to the fact that our objective was to carry out a review limited to the last 10 years, in a more specific field of DNA repair, the regulation of DSB repair genes (NHEJ and HR) by miRNAs. We are interested in this topic because of the therapeutic potential it represents.

  1. Following the guidelines for authors, citation should be as follows: 1. Author 1, A.B.; Author 2, Title of the article, Abbreviated Journal Name, Year, Volume, page range. Why did the authors of the review change that.

A: We appreciate the observation, which was addressed and is reflected in the current version of the manuscript.

Reviewer 2 Report

The Manuscript written by Ivan et al. explored the connection between microRNAs and DNA repair genes functioning in NHEJ and HR pathways. For the most part, the review reflects the current status of miRNA studies in NHEJ and HR, but certain literatures are ignored. A comprehensive search should be executed to ensure complete coverage of the literatures. In general, the manuscript is well-written, and I shall gladly recommend it be published in International Journal of Molecular Sciences following some revision.

Specific comments:

1 The authors claimed that they have included articles published between 2010-2021 focusing on miRNAs targeting DNA repair factors in NHEJ or HR pathway, but certain references have been missed. While Mir-502 targets both Ku70 and XLF, two essential factors functioning at different stages of NHEJ, such an important literature has been ignored. This should also be corrected in Figure 3, the legend of Figure 3 and in text.

Smolinska A., Swoboda J., Fendler W., Lerch M.M., Sendler M. and Moskwa P. (2020) MiR-502 is the first reported miRNA simultaneously targeting two components of the classical non-homologous end joining (C-NHEJ) in pancreatic cell lines. Heliyon 6, e03187

The following two literatures have been ignored.

Mir-130b on CTIP

Yang L., Yang B., Wang Y., Liu T., He Z., Zhao H. et al. (2019) The CTIP-mediated repair of TNF-α-induced DNA double-strand break was impaired by miR-130b in cervical cancer cell. Cell Biochem. Funct. 37, 534–544

miR-493-5p on MRE11

Meghani K., Fuchs W., Detappe A., Drané P., Gogola E., Rottenberg S. et al. (2018) Multifaceted impact of microRNA 493-5p on genome-stabilizing pathways induces platinum and PARP inhibitor resistance in BRCA2-mutated carcinomas. Cell Rep. 23, 100–111

2 Two Reference 37 and two reference 35 have been listed in the reference section, which has caused confusion. The authors need to thoroughly check the text where ref 37 was cited and discussed to accurately match text and citation.

3 Beside the 1st ref 37 (Hu B et al. J Biol Chem 292:3531-3540) has been withdrawal due to spliced immunoblot, therefore it should not appear in the current review article as a reference.

4 The description of HR pathway and the role of each components in HR is oversimplified. A two-step end resection is required for the generation of long single strand DNA 3’ overhang for invasion of double stranded DNA. MRN-CTIP are considered essential for the initionation of short single strand DNA overhang followed by either exo1 or DNA2-RPA to promote long end resection. This issue needs to be addressed in the revised review manuscript. MRE11 endonuclease activity is at least as important as its 3’ to 5’ exonuclease, the authors need to address this to prevent misunderstanding by the readers.

5 In the section of BRCA1 and BRCA2, it appears that the relationship between miRNAs and BRCA1 is complex. Both negative and positive regulatory roles have been described for different sets of miRNAs. It is critical for the authors to distinguish the role of indirect regulatory miRNA, from direct regulatory miRNAs, which binds 3’UTR or coding regions of BRCA1. Additionally, long non-coding RNAs sequestering miRNAs as a sponge have been mentioned in this review article. To reduce confusion, the section of BRCA1 and BRCA2 should be rearranged with subsection title.

6 in line 144, the ref supports a role of mir-1323 in suppression of DNAPK-cs is ref35, not ref36.

7 in line 409, the ref supports a role of mir-155 in suppression of POLD1 is ref109, not ref119.

Author Response

Reviewer 2

Dear reviewer, your comments are encouraging, and we thank you for the time spent reviewing the submitted manuscript. Undoubtedly, these will contribute to improve the revision work that we present. We regret the bibliography omissions we made, we have added 

Comments and Suggestions for Authors

The Manuscript written by Ivan et al. explored the connection between microRNAs and DNA repair genes functioning in NHEJ and HR pathways. For the most part, the review reflects the current status of miRNA studies in NHEJ and HR, but certain literatures are ignored. A comprehensive search should be executed to ensure complete coverage of the literatures. In general, the manuscript is well-written, and I shall gladly recommend it be published in International Journal of Molecular Sciences following some revision.

Specific comments:

1 The authors claimed that they have included articles published between 2010-2021 focusing on miRNAs targeting DNA repair factors in NHEJ or HR pathway, but certain references have been missed. While Mir-502 targets both Ku70 and XLF, two essential factors functioning at different stages of NHEJ, such an important literature has been ignored. This should also be corrected in Figure 3, the legend of Figure 3 and in text.

Smolinska A., Swoboda J., Fendler W., Lerch M.M., Sendler M. and Moskwa P. (2020) MiR-502 is the first reported miRNA simultaneously targeting two components of the classical non-homologous end joining (C-NHEJ) in pancreatic cell lines. Heliyon 6, e03187

The following two literatures have been ignored.

Mir-130b on CTIP

Yang L., Yang B., Wang Y., Liu T., He Z., Zhao H. et al. (2019) The CTIP-mediated repair of TNF-α-induced DNA double-strand break was impaired by miR-130b in cervical cancer cell. Cell Biochem. Funct. 37, 534–544

miR-493-5p on MRE11

Meghani K., Fuchs W., Detappe A., Drané P., Gogola E., Rottenberg S. et al. (2018) Multifaceted impact of microRNA 493-5p on genome-stabilizing pathways induces platinum and PARP inhibitor resistance in BRCA2-mutated carcinomas. Cell Rep. 23, 100–111

A: We appreciate your comment, in addition to including the references you indicate, we did the search again and, in addition to the works you mention, we found that an additional study that we have included (Srinivasan G., Williamsona, E., Konga, K., Jaiswala, A., et al (2019) MiR223-3p promotes synthetic lethality in BRCA1-deficient cancers PNAS, 116 (35): 17438-17443).

2 Two Reference 37 and two reference 35 have been listed in the reference section, which has caused confusion. The authors need to thoroughly check the text where ref 37 was cited and discussed to accurately match text and citation.

A: In the current version we have fixed the bibliography that was double cited.

3 Beside the 1st ref 37 (Hu B et al. J Biol Chem 292:3531-3540) has been withdrawal due to spliced immunoblot, therefore it should not appear in the current review article as a reference.

A: Not knowing the retraction of this study, Hu B et al. J Biol Chem 292:3531-3540, is that we had included it. You are right, we eliminate this reference in the actual version.

4 The description of HR pathway and the role of each components in HR is oversimplified. A two-step end resection is required for the generation of long single strand DNA 3’ overhang for invasion of double stranded DNA. MRN-CTIP are considered essential for the initionation of short single strand DNA overhang followed by either exo1 or DNA2-RPA to promote long end resection. This issue needs to be addressed in the revised review manuscript. MRE11 endonuclease activity is at least as important as its 3’ to 5’ exonuclease, the authors need to address this to prevent misunderstanding by the readers.

A: We agree that we overly summarize the HR pathway. Therefore, we add the information indicated in the text: "HRR requires invasion of homologous or sister chromatid strands to repair DNA end DSBs and is considered a typically error-free DNA repair mechanism [48] (Figure 4). A critical initial step in establishing this repair mechanism is DNA end resection, which generates a long 3' single-stranded DNA that can invade the homologous DNA strand.This step, in addition to blocking the entry of repair Ku proteins by NHEJ, promotes ATM and ATR activation and is therefore restricted to the late S phase and G2 phases of the cell cycle [48, 77] (Figure 4).DNA end resection initiation begins with MRN complex (MRE11, RAD50, and NBS1), SRCAP, and CtIP.It is important to mention that the dual activity of MRE11, functioning as both endo and exonuclease, is essential to carry out resection initiation After that, resection continues extension, in which SMARCAD1 cooperates with EXO1 and BLM/DNA2 Subsequently, the process continues when the ssDNA- and ends are covered with RPA. In a third step, RPA is replaced by RAD51 in a BRCA1/2-dependent process, to finally carry out the recombinase reaction using a homologous DNA template [49]. For a detailed description of HRR, there are more extensive reviews on the topic [8,49–52]. ".

5 In the section of BRCA1 and BRCA2, it appears that the relationship between miRNAs and BRCA1 is complex. Both negative and positive regulatory roles have been described for different sets of miRNAs. It is critical for the authors to distinguish the role of indirect regulatory miRNA, from direct regulatory miRNAs, which binds 3’UTR or coding regions of BRCA1. Additionally, long non-coding RNAs sequestering miRNAs as a sponge have been mentioned in this review article. To reduce confusion, the section of BRCA1 and BRCA2 should be rearranged with subsection title.

A: Indeed, in this review we found that BRCA1 is one of the genes that has shown more interactions with miRNAs. As you have well observed, by not specifying the type of interaction that occurs in said relationship, the text is confusing for the reader. In the new version, in addition to having specified the type of interaction in those cases where it was not clear enough, we also reorganized the information. We first describe interactions for BRCA1 and BRCA2, then the interactions of different kinds and including a new work on the negative regulation of miR223-3p and BRCA1 and finally the interactions involving BRCA2.

6 in line 144, the ref supports a role of mir-1323 in suppression of DNAPK-cs is ref35, not ref36.

A: We adjust all the references

7 in line 409, the ref supports a role of mir-155 in suppression of POLD1 is ref109, not ref119.

A: We already adjust all the references